# A Randomised Controlled Trial to Evaluate the Administration of the Health Improvement Card as a Health Promotion Tool: A Physiotherapist-Led Community-Based Initiative

**DOI:** 10.3390/ijerph17218065

**Published:** 2020-11-02

**Authors:** Yiwen Bai, Xubo Wu, Raymond CC Tsang, Ruisheng Yun, Yan Lu, Elizabeth Dean, Alice YM Jones

**Affiliations:** 1Department of Physical Therapy, Shanghai University of Traditional Chinese Medicine, Shanghai 201203, China; baiyiwen@shutcm.edu.cn (Y.B.); wuxubo@shutcm.edu.cn (X.W.); ruishengyun@pku.edu.cn (R.Y.); 2Department of Rehabilitation Medicine, Seventh People’s Hospital of Shanghai University of Traditional Chinese Medicine, Shanghai 200137, China; luyan_nini@126.com; 3Department of Physiotherapy, MacLehose Rehabilitation Centre, Hong Kong, China; tsangccr@ha.org.hk; 4Department of Physical Therapy, University of British Columbia, Vancouver, BC V6T 1Z3, Canada; elizabeth.dean@ubc.ca; 5School of Health and Rehabilitation Sciences, The University of Queensland, Brisbane 4072, Australia

**Keywords:** Health Improvement Card, Chinese community-dwelling adults, physiotherapy-led health education strategy

## Abstract

A randomised controlled trial was conducted to evaluate the administration of the Health Improvement Card (HIC) on lifestyle practices and biometric variables in community-dwelling Chinese participants. Adults living in Shanghai were randomly assigned to either the HIC-intervention or control group. Measurements/assessments were conducted at baseline and three-month follow-up. Supervised physiotherapy students administered the HIC and four standardised questionnaires related to health and wellbeing. Both groups received a health promotion education pamphlet. Based on participants’ HIC biometric and lifestyle scores, students prescribed lifestyle, and exercise advice to the HIC-intervention group. 171 individuals (39 men, 132 women) (mean age 68.4 ± 9.7 y) participated. At follow-up, body mass index (BMI) and waist circumference decreased significantly in the HIC-intervention group. Furthermore, the number of participants in the HIC-intervention group categorised as low risk regarding their physical activity and dietary practices, increased by 32.2% and 20%, respectively. Changes in standardised questionnaire scores did not meet minimum clinically importance differences in either group. This is the first study to demonstrate that HIC-informed health promotion education can improve people’s lifestyle practices, thereby, objective biometric variables. Evaluation of the effect of HIC-informed lifestyle education on some biometric parameters (blood pressure and BMI) may warrant a longer timeframe.

## 1. Introduction

Comparable to other developing countries, globalisation has led to rapid economic and social change in China which is contributing to unhealthy lifestyle practices and increased risk of non-communicable diseases (NCDs) within the population [1,2,3]. By 2030, 17% of the Chinese population is projected to be over 65 years of age with the prevalence of NCDs in China expected to triple [4]. In 2020, Shanghai has a population of 22.3 million, of which 32% are at least 60 years of age. In anticipation, the Shanghai provincial government prioritised the development of community health campaigns to promote population health and control of NCDs [5].

To increase the awareness of NCD risk factors by health professionals and the public, the World Health Professions Alliance designed the Health Improvement Card (HIC) for health professionals to readily assess common lifestyle-related attributes and NCD lifestyle-related risk factors within the clinical context [6]. The two-page HIC [7] consists of three principal sections. The first section includes patient information including age, height, weight, and waist circumference. The second section is the Biometric Scorecard which records four parameters, namely, body mass index (BMI), fasting blood sugar (FBS), cholesterol, and blood pressure. The third section is the Lifestyle Scorecard, which documents four modifiable lifestyle practices, namely, healthy diet (daily fruit and vegetable servings), daily physical activity (PA), tobacco use, and use of alcohol. The entries on both the Biometric Scorecard and the Lifestyle Scorecard are traffic-light colour coded, i.e., red for high risk, amber for moderate risk, and green for low risk. The colour-coded system readily highlights the level of risk for the benefit of both the health professional and the patient. Additional sections at the end of the biometric and lifestyle practices sections include the health improvement action plan, where the patient and health care professional jointly agree to the plan, stating their commitments to it, and specific actions to be implemented by each of them. A target date for goal achievement is set. 

Recently, the HIC has been translated into Chinese and integrated into the entry-level curriculum of the Department of Physical Therapy, Shanghai University of Traditional Chinese Medicine [8]. In the HIC Chinese translation study, physiotherapy students reported being receptive and confident in administering the HIC to assess an adult’s health attributes and lifestyle behaviours, and prescribe a lifestyle behaviour change programme based on that assessment [8]. Although the HIC was developed for purposes of health promotion, there are no studies of either its effectiveness in modifying lifestyle practices, in turn, biometric health indices; or evaluation of it, as a routine clinical tool used by health professionals. 

This study aimed to examine the effectiveness of administration of the HIC in effecting positive lifestyle behaviour change related to reducing NCD risk factors, in turn, optimising markers of good health reflected in the biometric data. We hypothesised that systematic application of the HIC is associated with changes in lifestyle practices, in turn, favourable changes in biometric markers over time. Further, we anticipated that physiotherapy students who had been formally instructed in the clinical application of the HIC as an assessment and education tool would report that administering it to people in the community, constituted a positive learning experience. 

## 2. Materials and Methods 

### 2.1. Study Design

We conducted a randomised controlled trial involving three community centres in Shanghai that are affiliated with the Seventh People’s Hospital. Participants were enrolled between 1 January and 30 March 2019. This study was registered at the ChineseClinicalTrials.gov site (Identifier: ChiCTR1900025616). Approval to conduct the study was granted by the Ethics Committee, the Seventh People’s Hospital of Shanghai University of Traditional Chinese Medicine (Ethics approval number: 2018-IRBQY-013). The protocol for this study was registered at Protocol.io (https://dx.doi.org/10.17504/protocols.io.bhz3j78n). Written informed consent was obtained from each participant prior to study participation. 

### 2.2. Participants

Adults between 50 and 90 years of age were recruited through pamphlets, posters, and information sessions distributed and organised by the three participating community centres. Inclusion criteria were Shanghai residents, no observable communication barrier, and no hospital admission within three months prior to the study. Exclusion criteria were also based on self-reporting: mental illness, dementia or observed communication difficulty potentially interfering with participation, heart disease (e.g., cardiac dysrhythmias, heart failure, post-infarction angina, or a history of cardiogenic shock), liver dysfunction, renal dysfunction, and malignancy.

### 2.3. Sample Size

There has been no previous use of the HIC that can serve as a basis for estimating sample size. Therefore, the hypothesised difference in BMI change (as the primary outcome) between the intervention and control groups was predicted to be small to moderate, corresponding to a Cohen effect size between 0.2 (small) to 0.5 (medium). Based on a conservative hypothesised effect size of 0.4 in testing the mean difference of BMI change scores between the HIC-intervention group and control group with two-tailed independent t test using the G * Power 3.1.9.4 (Universität Kiel, Germany), with alpha set at 0.05 and a statistical power of 0.8; a total of 200 participants were required with 100 participants in each of the HIC-intervention group and control group.

### 2.4. Procedures

Consenting participants eligible for the study were randomly allocated to the HIC-intervention group and the control group. Group randomisation was based on a computer-generated sequence list. The randomisation list was concealed in sequentially numbered, sealed, opaque envelopes and prepared by an independent physiotherapist not involved in subject recruitment. Each new participant was assigned a number sequentially, and the corresponding envelope was opened to determine their group assignment. The procedures were comparable across the three participating community centres.

### 2.5. Dependent Variables

Dependent variables were measured or assessed at baseline and at three-month follow-up. First, the Chinese version of the HIC [8] was administered to participants by assigned supervised physiotherapy students. As part of the HIC, the following data were collected for each participant: (1) demographic data including gender, age, measured height, weight, and waist circumference (WC); (2) biometric data calculated body mass index (BMI), including random rather than fasting blood sugar (RBS), total cholesterol (TC), and systolic and diastolic blood pressures (SBP and DBP). A single, qualified nurse who was blinded to participant group allocation was invited to assist with recording the biometric variables for all involved participants at each community centre, as physiotherapists are not certified to take blood samples. In addition, four lifestyle practices were categorised based on the criteria specified on the HIC for low, moderate, and high NCD risk. These are healthy diet (daily fruit and vegetable consumption); PA (minutes/day); tobacco smoking (yes or no); and alcohol use (daily amount). 

Positive emotions have been linked to lower risk of specific diseases [9]. To determine the degree to which a lifestyle education intervention might impact other aspects of the health and wellbeing of participants, participants completed the validated Chinese version of the following questionnaires: General Anxiety Disorders-7 (GAD-7) [10] for assessment of anxiety level; Patient Health Questionnaire-9 (PHQ-9) [11] for assessment of depression; Pittsburgh Sleep Quality Index (PSQI) [12] for assessment of sleep quality; and Medical Outcome Study Short Form 36 (SF-36), a health-related quality of life questionnaire [13]. Assistance completing the questionnaires was provided to illiterate participants (11 in the HIC-intervention group and 6 in the control group) by the physiotherapy students.

### 2.6. Independent Variable

Twenty fourth-year physiotherapy students who received formal didactic and practical instruction in administering the HIC during their third year of study, as described by Wu et al. [8], volunteered to participate. The study and its requirements were explained to the students by the investigators and signed consent was obtained. Each student was assigned to follow 10 participants over the three participating community centres. Data collection was conducted on Saturdays and Sundays. Two qualified physiotherapists supervised the same students and were present at all times. Their role was to serve as a quality control check for the assessments and lifestyle education prescribed to participants by the physiotherapy students. 

Participants in the HIC-intervention group attended a discussion session with each of their assigned physiotherapy student. Based on data from the HIC, participants received tailored lifestyle education in the form of general lifestyle advice. Examples include, smoking cessation strategies, a reduction in alcohol consumption, and a healthy whole-food, plant-based diet with a low sodium and cholesterol content was recommended. Specifically, participants were advised to consume 500 g of vegetables and fruit per day, limit salt intake to less than a teaspoon, soy sauce to less than half a Chinese spoon, and avoid pickles and salted duck eggs. 

As indicated, participants were prescribed an individual exercise programme with mode, frequency, and intensity of exercise. They were encouraged to develop an action plan based on lifestyle behavioural goals targeted at achieving a lower risk score for the components of the Biometric Scorecard and the Lifestyle Scorecard of the HIC. A frequently recommended exercise program was to walk around the community centre for 30 min each day, 3 times per week. The instruction to indicate exercise intensity was: “it is normal to sweat a little and your breathing rate should increase with walking. If you do not sweat or have no change in the breathing rate, you should walk at a faster pace.” 

All participants were given a standard brochure on healthy lifestyle practices and lifestyle behaviour change published by the Shanghai Seventh People’s Hospital. It provides basic information on the benefits of consuming a healthy diet, engaging in optimal PA for health, weight control, sleep hygiene, smoking cessation, and alcohol moderation. Finally, participants in the HIC-intervention group were invited to return to their respective community centres to meet with the same physiotherapy student monthly for follow-up and potential modification of the lifestyle goals and action plan. The control group had no other intervention and returned only for the three-month follow-up. 

Three months after baseline, participants from both groups were invited to be re-assessed at their community centres. Comparable to the baseline assessment, the biometric parameters and lifestyle practices as per the HIC were repeated in a comparable manner to baseline, as were the health-related standardised questionnaires. 

### 2.7. Equipment and Its Reliability

A glucometer (ACCU-Check Performa, SN79527019132, Roche, Mannheim, Germany) was used for measurement of blood sugar; the “Total Cholesterol Tester” (TCT) (Bene Check, BKM 13–1, General Life Biotechnology Co., Ltd., Taipei, Taiwan) was used for measurement of total blood cholesterol; blood pressures were measured in a standardised manner with a sphygmomanometer (Shanghai Yutu, XJ11D, Shanghai, China). The reliability of the glucometer and TCT was tested by comparing samples taken from the right middle finger for the glucometer, and TCT with blood samples drawn from the right arm of 10 generally healthy individuals. Blood samples were then sent to the laboratory for analysis.

### 2.8. Student Feedback on Clinical Application of the HIC

Feedback from participating students about their experience with lifestyle education in the community was obtained using a modified, previously established questionnaire [8]. They were asked to rate their experiences based on twelve closed-ended questions using a four-point Likert rating scale (i.e., strongly agree, agree, disagree, and strongly disagree). Open-ended comments on how the use of HIC could further be improved were also sought. The first seven of these twelve questions were from our previously published questionnaire [8]. 

### 2.9. Statistical Analyses

Statistical analyses were performed using the IBM SPSS Statistics for Windows, Version 21.0 (Armonk, NY, USA: IBM Corp). Summary statistics for the demographic characteristics and other outcome variables were computed; means and standard deviations for continuous data and percentages for categorical data calculated. Differences between baseline and three-month follow-up for the outcome score variables were computed as “improvement” scores. Factorial ANOVA comparing the between-group improvement scores was conducted for all outcome measures with group and gender as independent variables. If there were no significant main effect of gender and interaction effect of group x gender in the factorial ANOVA, independent t tests were used to compare the improvement scores between the HIC-intervention and control groups, provided the improvement scores met the statistical assumption of approximating normality [14]. The 95% confidence intervals of mean differences of between-group improvement scores of all outcome variables were calculated. Mean differences between scores over time for each individual were compared with paired t-tests. The changes in “diet”, “physical activity”, and both “diet and physical activity” practices from high- to low-risk levels were analysed with McNemar tests. Where appropriate, alpha was set at 0.05.

## 3. Results

Two hundred adults provided consent to participate and were randomised into either the HIC-intervention group (*n* = 100) or control group (*n* = 100). There were no between-group differences in either demographic characteristics or baseline measurements (Table 1). 

At three-month follow-up, 171 participants (90 in the HIC-intervention group and 81 in the control group) were re-assessed. Reasons that participants gave for non-attendance are shown in the consort flow diagram (Figure 1). The diagram shows that of the 237 individuals assessed for eligibility, 100 were randomised to the HIC-intervention group and 100 to the control group. At the three-month follow-up, 10 and 19 participants were lost from the HIC-intervention group and the control group, respectively. 

### 3.1. Biometric Measure

Biometric measures at baseline and three-month follow-up are shown in Table 2 for both groups. At follow-up, BMI was reduced from 25.21 ± 3.42 to 24.82 ± 3.39 kg/m^2^ and WC was reduced from 91.52 ± 10.18 to 89.95 ± 9.75 cm for the HIC-intervention group (*p* < 0.05), whereas there was no change in either variable, in the control group. Moreover, DBP decreased from 80.80 ± 8.18 to 77.20 ± 10.04 mm Hg (*p* < 0.05) in the HIC-intervention group and no change was observed in the control group. However, when the two groups were compared, there were no differences for the biometric variables, i.e., BMI, blood sugar, cholesterol, and blood pressure. While there appears to be a large disparity in gender in our cohort, there was no significant gender and intervention interaction effect (gender x group) on the improvement scores in any of the outcome variables.

### 3.2. Changes in Lifestyle Behaviour

Table 3 shows the number of participants at various risk levels for each biometric and lifestyle behaviour variable. Most participants were low risk tobacco and alcohol users. 

Subgroup analysis showed that the number of participants in the HIC-intervention group meeting the low-risk level criterion for PA has increased by more than 30% (from 55.6% to 87.8% in the low-risk category), and the number meeting the healthy diet criterion increased by 20%. The number of participants who were low risk for both PA and diet, increased by 28.9% (Table 4). In the control group, however, the number of participants categorised as low risk regarding PA or diet, or both, did not differ from baseline (Table 4). 

### 3.3. Changes in Health and Well Being Questionnaire Scores

A within HIC-intervention group and a between-group difference in change score for PHQ-9 were observed (both *p* < 0.05). There was a within-group improvement in PSQI (*p* = 0.027) and SF36 PCS (*p* = 0.045) in the control group (Table 5), but there was no between-group difference in the “change” scores. None of the changes in the GAD-7, PHQ-9, PSQI, and SF-36 scores reached a minimum clinically important difference for any respective score (Table 6). 

### 3.4. Students Responses 

Responses of the physiotherapy students regarding the administration of the HIC to community-dwellers to promote health, and their confidence in guiding participants to adopt healthy lifestyle practices are shown in Table 7. Compared with previously published data on Chinese physiotherapy students’ views regarding their understanding of the role of HIC in health promotion education [8], students involved in the present study reported an even greater understanding of the role of the HIC and greater confidence in interpreting the HIC data and progressing participants, based on their goals and action plans. Students either strongly agreed, or agreed, that they gained confidence in providing advice to their patients about the lifestyle practices included in the HIC and that active participation in this study facilitated their understanding of the role of a physiotherapist in community and public health (Table 7). Open-ended comments were received from six students. Their recommendations included use of an electronic version of the HIC and inclusion of pictures of appropriate portions of fruit and vegetables, and alcohol consumption. One student questioned the accuracy of one participant’s report of tobacco use and what measures could be taken to maximise response validity. 

## 4. Discussion

To our knowledge, this is the first study to examine outcomes of administering the HIC with respect to changing lifestyle practices, in turn, effecting objective biometric outcomes. Compared with the control group, findings in the HIC-intervention group support the value of the HIC in health and lifestyle education to change lifestyle practices, in turn, potential objective metrics, e.g., reduction in BMI, WC, and DBP. However, there were no observable differences in the between-group changes in biometric markers of risk for NCDs. We postulate that follow-up at three months may have been an insufficient timeframe to detect objective evidence of changes. 

Although we recruited 200 participants based on sample size calculation, 29 were lost at three-month follow-up. Some biometric markers can change rapidly in response to changes in diet and PA; however, others require a long-term commitment. There is strong evidence to support the relationship between high levels of PA and weight loss [15], but effective weight loss programmes require several months [16]. Similarly, blood sugar and cholesterol levels take time to respond to dietary changes. 

The HIC was designed, based on best evidence, to inform and guide NCD risk reduction globally, with tailored advice about specific healthy lifestyle practices based on objective measures [6]. Before objective evidence is apparent, modification of lifestyle practices needs to be adopted and consistently adhered to. In the HIC-intervention group, the number of participants meeting the low risk or green colour-coded criterion increased in two of the modifiable behaviours associated with NCD risk, namely PA and diet. Further, the number that achieved this criterion for both PA and diet, increased from 18 to 47%. This suggests that the health advice provided by our team including physiotherapy students and qualified supervising physiotherapists, appeared to be effective in positively modifying lifestyle behaviours with participants reducing their BMIs and WCs. This is not surprising, as the interaction between PA and BMI is well established [17,18]. Although the changes in BMIs and WCs were relatively small, the improvement in our intervention group was positive and statistically significant. Conversely, all changes in the control group were non-significant. This suggests that health education intervention demonstrates a trend in a positive direction with respect to improving health, should the positive health practices be sustained. 

Participating students reported being confident and comfortable administering the HIC as a basis for communicating about lifestyle behaviour change education with participants. Administration of the HIC provided an effective and practical instrument to initiate systematic examination of participants’ NCD risk factors. The HIC not only guided advice specific to each individual’s particular needs but also provided a meaningful, quantifiable tool to gauge the progress in lifestyle behaviour modification. Students did not report challenges with respect to explaining and administering the HIC with any participant, irrespective of gender and age, suggesting the HIC is user-friendly and readily understood by the lay public.

Given that smoking cessation and alcohol moderation programmes have not been formally established in the participating community centres, the students were unable to modify tobacco use or excessive alcohol consumption beyond general advice regarding the dangers of these practices and benefits of quitting smoking and limiting alcohol consumption. However, the number of participants who allegedly smoked was small and alcohol use was reported to be negligible. Nonetheless, reinforcement of these positive lifestyle behaviours was of value not only to the participant, but also potentially the participant’s family members. 

Changes we observed in the scores from the GAD-7, PHQ-9, PSQI, and SF-36 physical component score (PCS), and SF-36 mental component score (MCS) were less than minimal clinically important differences reported for the scores for the respective questionnaires [19,20,21,22]. It is interesting to note an apparent increase in the PHQ-9 score in the HIC-intervention group, whereas a decrease was observed in the control group. More HIC participants shifted from the “minimally” depressed category to the “mildly” depressed category. Perhaps these participants were less successful compared with others in achieving low-risk categories for their lifestyle practices at three-month follow-up, thus contributed to the mild depressed feeling. Most participants reported minimal general anxiety, however some 15–21% in both groups reported “fairly bad” sleep quality. Sleep quality and depression are common in the elderly [23]. While the analysis of sleep quality in our participants was beyond the scope of our study, these findings suggest that strategies to target sleep quality may be indicated given poor sleep hygiene is an independent correlate of NCD risk. 

Specifically, our study examined the effect of administering the HIC by supervised physiotherapy students as a clinical health promotion tool to assess the need for and provide lifestyle advice to Chinese community-dwelling adults. The physiotherapy profession has been urged to assume a leadership role in the promotion of the long-term benefits of effective, sustained, non-pharmacological lifestyle change to minimise the risk of NCDs [24]. Although the practice of physiotherapy particularly at an international standard is relatively new to China [25], China appears to be the first country to report the inclusion of the HIC into some of the curricula of its physiotherapy programmes [8]. Students involved in the current study were part of the cohort that was first formally introduced to the HIC in their curriculum [8]. With the addition of a few questions, a similar feedback questionnaire was used to gauge the students’ receptivity to administering the HIC as a clinical tool. Comparing the responses from students’ experiences in this study and our previous one, we observed that 90% of students in the present study, compared with 54%, reported [8] they strongly agreed that they understood the purpose and role of the HIC in clinical practice. Further, 70% of students in this current study compared to 20% previously, strongly agreed with the statement that, “they could interpret the results and/or progress a patient using the HIC in an appropriate manner”. In addition, in the present study, students were asked to rate their confidence to recommend necessary “actions” based on the data obtained the HIC; setting healthy lifestyle targets for a participant; and better understanding the role of a physiotherapist in community and public health. Overall, the students in the study were positive. Open-ended comments from students in this and our previous study [8] identified uncertainty about what constitutes an appropriate daily portion of fruits and vegetables per day. Although this is described in the health professionals’ guide to the HIC, the students’ repeated response reinforced the need to make this more explicit to health professionals and patients alike. 

Over several decades, the NCDs have become a socioeconomic burden globally including China. China has embarked on a long-term health campaign called “Healthy China 2030” [26] aimed at increasing awareness and enhancing the population’s general health by promoting healthy lifestyles, optimising health services, improving health security, building a healthy environment, and developing health industries. By being involved in community health and confirming their competencies to modify lifestyle practices, specifically through appropriate PA and dietary recommendations, physiotherapists can minimise NCD risk, thereby support the “Healthy China 2030” campaign by expanding their role in community health education and health promotion.

### 4.1. Limitations

Although our sample size met the requirements, a larger sample size may have increased effect sizes of the HIC-informed health promotion intervention as well as adjusted for participants lost at follow-up. With respect to research participation, volunteering for research in China is uncommon, which may have limited expressions of interest by the public. Second, only random blood glucose rather than fasting blood sugar measurement was available through the participating community centres. Comparisons between baseline and three-month follow-up random blood sugar levels were therefore not entirely meaningful. Finally, a three-month follow-up may be too short a timeframe to demonstrate objective changes in biometric variables following effective change in lifestyle practices. Many clinicians including physiotherapists may not have the advantage of following patients over several months. Despite this, the evidence is unequivocal that supporting and guiding patients toward healthy lifestyle practices is the first step toward their adopting healthier lifestyles and effecting change in objective measures of NCD risk factors; a value consistent with the long-standing goals of the World Health Organisation [1]. 

Of relevance is the potential impact of sex, age, and education of the participants on outcomes of use of the HIC in the community and clinical settings in China. However, in-depth examination of the effect of these variables is beyond the scope of this study. This is a focus of future studies. 

Health professionals need to practice as a team, so that their messaging to patients regarding healthy lifestyle practices is consistent. The HIC was developed by the five leading established health professions in the world; thus, use of such a standard tool across health professions, namely, medicine, nursing, pharmacy, and dentistry, as well as physiotherapy would maximise the effectiveness of lifestyle behaviour change across individuals and time, and globally. 

### 4.2. Utility of the HIC and Future Studies

Our findings support the contention that the HIC has the potential for readily assessing and initiating health education in adults, which may impact clinical disease. Given the unequivocal importance of NCD risk factor assessment in every patient and that education on lifestyle behaviour changes should be tailored to individuals, expanding the HIC to include other well recognised NCD risk factors needs to be considered. We thus propose that in addition to smoking, diet, activity, and alcohol consumption important contributors to NCD risks such as sleep hygiene and general mental health including anxiety and stress levels warrant inclusion in future. However, the HIC was purpose designed into two pages to facilitate clinician completion and encourage clinical compliance. A longer more complex version could be considered when clinicians become more experienced with its use.

Finally, future studies need to replicate and extend this work so that the effects of other variables such as sex, age, and education can be examined. Another subsequent step is implementing its use in the clinical setting and for varying patient populations and ages. Quantitative studies such as this are needed, but qualitative studies also need to be undertaken to identify facilitators and barriers to the use of the HIC by clinicians, and what modifications might facilitate the learning experience for patients. 

## 5. Conclusions 

To our knowledge, this the first study to report the outcome of administering the HIC as a clinical tool for tailoring lifestyle education, i.e., changing lifestyle behavioural practices, in turn, biometric variables associated with NCD risk. Further, this was the first such trial carried out in China. Our findings supported that administration of the HIC by physiotherapy students under supervision was associated with favourable effects on participants’ lifestyle practices, e.g., increased PA and healthier diet, in turn, their BMIs. Timeframes longer than three months are likely needed to effect substantial changes in objective markers and health-related indices such as health-related quality of life, sleep, and mental wellbeing. Further studies are needed to promote the uptake and routine adoption of the HIC by established health professionals, educators and students across settings and countries globally, given that once individuals change their lifestyle practices, the likelihood of long-term objective change is high. Finally, our findings support the potential of physiotherapists to actively engage in NCD risk factor prevention and reduction at the clinical level, in the interests of supporting national health campaigns.

## Figures and Tables

**Figure 1 ijerph-17-08065-f001:**
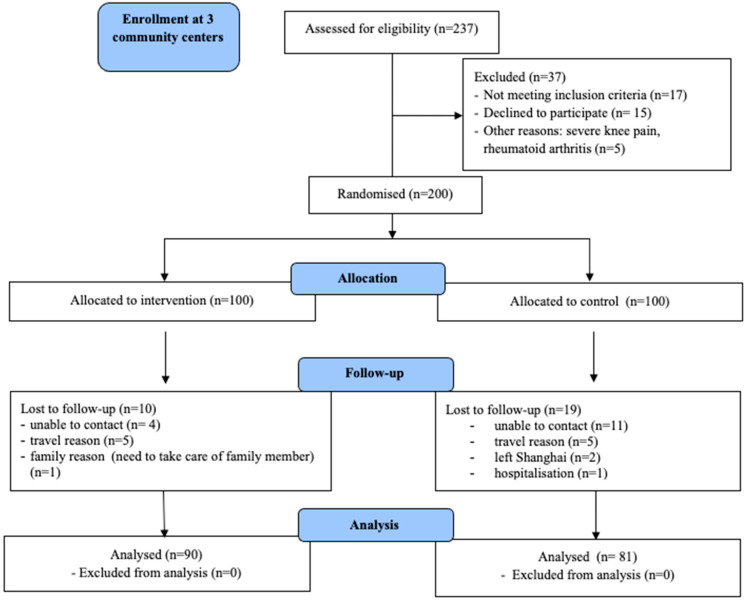
Consort Flow Diagram.

**Table 1 ijerph-17-08065-t001:** Baseline characteristics of participants in the intervention and the control groups. Data are number (%) unless otherwise stated.

Characteristic	HIC- Intervention Group	Control Group	*p* Value
	(*n* = 90)	(*n* = 81)	
Sex, (men/women)	17/73	22/59	0.20 *
Age, y (Mean ± SD)	69.5 ± 8.5	67.1 ± 10.7	0.16 #
50–59	7 (7.8)	8 (9.9)	
60–69	44 (48.9)	40 (49.4)	
70–79	26 (28.9)	28 (34.6)	
80–89	13 (14.4)	5 (6.2)	
Education			0.72 *
Illiterate	11 (12.2)	6 (7.4)	
Primary school	17 (18.9)	18 (22.2)	
Junior high	52 (57.8)	47 (58)	
Secondary school	7 (7.8)	8 (9.9)	
Post-secondary	2 (2.2)	1 (1.2)	
Smoker	13 (14.4)	15 (18.5)	0.43 *
Alcohol (> 1glass/day)	13 (14.4)	15 (18.5)	0.76 *
Hypertension *	31 (34.4%)	34 (41.9%)	0.40 *
Diabetes *	13 (14.4%)	4 (17.3%)	0.61 *

* Chi-square test; # Independent t-test; HIC—Health Improvement Card.

**Table 2 ijerph-17-08065-t002:** Between group comparisons of the changes in biometric variable scores at three-month follow-up.

Outcome	Time	HIC-Intervention Group (*n* = 90)		Control Group (*n* = 81)		Mean Difference of Between-Group Change Scores (95% CI) (*p* Value) #
		Mean Score (SD)	Mean Change from Baseline to Follow-up (SD) (*p* Value) *	Mean Score (SD)	Mean Change from Baseline to Follow-up (SD) (*p* Value) *	
BMI (kg/m^2^)	Baseline	25.21 (3.42)	0.40 (0.85) (< 0.001)	24.99 (2.99)	0.12 (1.14) (0.353)	0.28 (−0.02 to 0.58) (0.069)
	Follow-up	24.82 (3.39)		24.88 (2.92)		
WC (cm)	Baseline	91.52 (10.18)	1.57 (6.51) (0.024)	91.81 (9.00)	0.70 (6.45) (0.335)	0.88 (−1.08 to 2.84) (0.379)
	Follow-up	89.95 (9.75)		91.11 (7.64)		
RBS (mmol/L)	Baseline Follow-up	7.86 (3.61) 7.92 (3.28)	−0.06 (3.68) (0.874)	7.02 (2.17) 6.87 (1.80)	0.15 (2.13) (0.527)	−0.21 (−1.13 to 0.71) (0.650)
TC (mmol/l)	Baseline	4.26 (1.08)	−0.18 (1.16) (0.145)	4.01 (0.95)	−0.12 (1.03) (0.308)	−0.06 (−0.40 to 0.27) (0.718)
	Follow-up	4.44 (1.07)		4.13 (0.85)		
SBP (mmHg)	Baseline	132.80 (14.15)	0.99 (14.74) (0.526)	132.74 (12.89)	2.25 (16.98) (0.237)	−1.26 (−6.05 to 3.53) (0.605)
	Follow-up	131.81 (12.99)		130.49 (14.56)		
DBP (mmHg)	Baseline	80.80 (8.18)	3.60 (9.98) (0.001)	80.75 (7.93)	1.69 (12.16) (0.214)	1.91 (−1.44 to 5.25) (0.262)
	Follow-up	77.20 (10.04)		79.06 (11.76)		

* dependent t test; # independent t test; BMI, body mass index; WC, waist circumference; RBS, random blood sugar; TC, total cholesterol; SBP, systolic blood pressure; DBP, diastolic blood pressure.

**Table 3 ijerph-17-08065-t003:** Number of participants at various risk levels (high, moderate, and low) for specific biometric variables and lifestyle practices at baseline and three-month follow-up.

Parameters	HIC-Intervention Group (*n* = 90)		Control Group (*n* = 81)	
	Baseline	Follow-up	Baseline	Follow-up
	High|Medium|Low	High|Medium|Low	High|Medium|Low	High|Medium|Low
Body mass index	9|37|44	8|34|48	5|34|42	6|37|38
Random blood sugar	38|37|15	40|37|13	31|31|19	32|33|16
Total cholesterol	5|15|70	6|13|71	3|6|72	3|6|72
Blood pressure	31|45|14	25|46|19	42|23|15	44|21|16
Healthy diet	4|58|28	0|44|46	4|43|34	3|39|39
Physical activity	12|28|50	4|7|79	9|39|33	8|38|35
Tobacco use	5|85 (High|Low)	5|85 (High|Low)	9|72 (High|Low)	9|72 (High|Low)
Alcohol use	0|1|89	0|1|89	4|2|75	4|2|75

**Table 4 ijerph-17-08065-t004:** Changes in physical activity and dietary practices at three-month follow-up. Data are number of participants who achieved the low risk category for physical activity and diet. * McNemar test.

Lifestyle Practice	HIC-Intervention Group				Control Group			
	Baseline	Follow-up	Change Number (%) (95%CI)	*p* Value *	Baseline	Follow-up	Change Number (%) (95%CI)	*p* Value *
Physical Activity	50 (55.6%)	79 (87.8%)	29 (32.2%) (19.6–44.9%)	<0.001	33 (40.7%)	35 (43.2%)	2 (2.5%) (−11.6 to 16.6%)	0.864
Diet	28 (31.1%)	46 (51.1%)	18 (20.0%) (10.7–29.3%)	<0.001	34 (42.0%)	39 (48.1%)	5 (6.2%) (−1.0 to 13.3%)	0.180
Physical Activity + Diet	16 (17.8%)	42 (46.7%)	26 (28.9%) (18.1–39.7%)	<0.001	12 (14.8%)	20 (24.7%)	8 (9.9%) (0.4–19.3%)	0.077

**Table 5 ijerph-17-08065-t005:** Between group comparisons of the changes in scores from the standardised questionnaires at three-month follow up.

Out-come	Time	HIC-Intervention Group (*n* = 90)		Control Group (*n* = 81)		Mean Difference of Between-Group Change Scores (95% CI) (*p* Value) #
		Mean Score (SD)	Mean Change from Baseline to Follow-up (SD) (*p* Value) *	Mean Score (SD)	Mean Change from Baseline to Follow-up (SD) (*p* Value) *	
GAD-7	Baseline	1.14 (2.26)	−0.02 (1.82) (0.908)	1.63 (3.23)	0.30 (2.38) (0.267)	−0.32 (−0.97 to 0.33) (0.332)
	Follow-up	1.17 (1.98)		1.33 (3.04)		
PHQ-9	Baseline	1.72 (2.67)	−0.80 (2.22) (0.001)	2.42 (3.44)	0.14 (3.06) (0.400)	−0.94 (−1.74 to −0.13) (0.022)
	Follow-up	2.52 (2.97)		2.28 (2.96)		
PSQI	Baseline	7.84 (4.29)	0.16 (3.83) (0.701)	8.28 (4.61)	0.93 (3.70) (0.027)	−0.77 (−1.91 to 0.37) (0.184)
	Follow-up	7.69 (4.70)		7.36 (4.86)		
SF-36 PCS	Baseline	82.74 (15.52)	2.45 (13.96) (0.100)	80.70 (16.64)	3.97 (17.54) (0.045)	−1.52 (−6.29 to 3.24) (0.529)
	Follow-up	85.18 (12.80)		84.67 (14.34)		
SF-36 MCS	Baseline	90.09 (11.30)	−0.45 (12.34) (0.732)	86.33 (16.60)	1.03 (14.65) (0.530)	−1.47 (−5.55 to 2.60) (0.476)
	Follow-up	89.64 (10.71)		87.36 (13.37)		

* dependent t test; # independent t test; GAD-7, General Anxiety Disorders-7; PHQ-9, Patient Health Questionnaire-9; PSQI, Pittsburgh Sleep Quality Index, SF-36: The Medical Outcome Study Short Form 36, PCS: Physical Component Summary; MCS, Mental Component Summary.

**Table 6 ijerph-17-08065-t006:** Distribution of participants at various categories of anxiety, depression, and sleep quality at baseline and at three-month follow-up. Data are number (%).

Mental Health Parameters	Outcome	HIC-Intervention Group (*n* = 90)		Control Group (*n* = 81)	
		Baseline	Follow-up	Baseline	Follow-up
Anxiety Severity GAD-7 Scale Score	Minimal, 0–4	84 (93.4)	82 (91.1)	74 (91.4)	74 (91.4)
	Mild, 5–9 Moderate, 10–14 Severe, 16–21	4 (4.4)	8 (8.9)	4 (4.9)	5 (6.2)
		2 (2.2)	0 (0)	2 (2.5)	1 (1.2)
		0 (0)	0 (0)	1 (1.2)	1 (1.2)
Depression Severity PHQ-9 Scale Score	Minimal, 0–4	80 (88.9)	74 (82.2)	69 (85.3)	71 (87.7)
	Mild, 5–9	9 (10.0)	13 (14.5)	7 (8.6)	7 (8.6)
	Moderate, 10–14	0 (0)	2 (2.2)	4 (4.9)	2 (2.5)
	Moderately severe, 15–19	1 (1.1)	1 (1.1)	0 (0)	1 (1.2)
	Severe, 20–27	0 (0)	0 (0)	1 (1.2)	0 (0)
Sleep Quality PSQI Scale Score	Very good, 0–5	30 (33.3)	35 (38.9)	24 (29.6)	36 (44.4)
	Fairly good, 6–10	34 (37.8)	29 (32.2)	37 (45.7)	25 (30.9)
	Fairly bad, 11–15	22 (24.5)	19 (21.1)	11 (13.6)	12 (14.8)
	Very Bad, 16–21	4 (4.4)	7 (7.8)	9 (11.1)	8 (9.9)

GAD-7: General Anxiety Disorders-7; PHQ-9: Patient Health Questionnaire-9; PSQI: Pittsburgh Sleep Quality Index.

**Table 7 ijerph-17-08065-t007:** Feedback of physiotherapy students on the application of the HIC to community-dwelling adults. * SA, Strongly agree; A, Agree; D, Disagree; SD, Strongly disagree.

	Statements	SA *	A	D	SD
1	Physiotherapists should introduce the HIC to the general public	17 (85%)	3 (15%)	0	0
2	I understand the purpose and role of the HIC	18 (90%)	2 (10%)	0	0
3	I can provide advice to my patients about the actions prescribed on the HIC	12 (60%)	8 (40%)	0	0
4	I can identify instances where using the HIC would improve patient outcomes	11 (55%)	9 (45%)	0	0
5	I can justify my reasoning for choosing to implement the HIC with my patients	10 (50%)	10 (50%)	0	0
6	I understand when using the HIC may NOT be appropriate for a particular patient	6 (30%)	14 (70%)	0	0
7	I can interpret the results and/or progress a patient using the HIC in an appropriate manner	14 (70%)	6 (30%)	0	0
8	I have confidence in providing advice to my patients about the actions prescribed on the HIC	14 (70%)	6 (30%)	0	0
9	It is useful to my learning to have the opportunity to follow up the same patient each month	17(85%)	3 (15%)	0	0
10	I found the HIC to be a useful tool for me to work with the patient	15 (75%)	5 (25%)	0	0
11	The HIC makes it easy for me to set healthy lifestyle targets for the patient	14 (70%)	6 (30%)	0	0
12	Participation in this HIC project has given me some understanding of the physiotherapist’s role in the community	19 (95%)	1 (5%)	0	0

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
