# Peer review of "A Randomised Controlled Trial to Evaluate the Administration of the Health Improvement Card as a Health Promotion Tool: A Physiotherapist-Led Community-Based Initiative"

_ijerph, 2020, doi:10.3390/ijerph17218065_

Round 1

Reviewer 1 Report

The overall manuscript was well written and presented clear and concise results in the tables. The conclusion are supported by the results.

However, it was unclear as to why the researchers measured anxiety to determine the impact level of the lifestyle education intervention. Further details may be needed for explanation use of the assessment (line 125).  

Author Response

We are grateful for the reviewers’ comments which assisted us to improve the quality of the presentation of our manuscript. Our responses are shown below as AU in italics. In addition, edits are highlighted in yellow in the manuscript.

Reviewer 1

The overall manuscript was well written and presented clear and concise results in the tables. The conclusions are supported by the results.

  • However, it was unclear as to why the researchers measure anxiety to determine the impact level of the lifestyle education intervention. Further details may be needed for explanation use of the assessment (line 125).

AU: There is empirical evidence suggesting an association between a positive psychological status and health outcomes. We therefore thought it was prudent to try and explore whether our lifestyle education program had an impact on health and wellbeing. We have included a reference which we hope serves to illustrate our rationale for including an assessment of anxiety in our pilot study. Please refer to line 125.

Reviewer 2 Report

In this work, the evaluation of the administration of the Health Improvement Card through a randomized controlled card is presented, where Chinese participants are considered. Results demonstrate (in a time window of three months) that HIC-informed health promotion education can improve people’s lifestyle practices. In general, the manuscript is well-written and the topic is interesting; yet, some issues have to be addressed in order to clarify the contribution and applicability for the obtained results.

Why is the time window of three months used? Is it enough to obtain reliable results? Can you provide the evolution of the improvements for the month, i.e., for the first month, the second month, the third month (maybe a trend can be observed)?  

Please justify the values chosen in section 2.3. What happens with other values?

In the analyzed database, 171 individuals (39 men and 132 women) are considered. Can your results be negatively affected due to the disparity of gender?

Figure 1 needs a wider description.

In section 3.1, some reductions are for BMI and WC, but the positive impact of such reductions is not discussed.

Although general quantitative results are presented, a deeper discussion is desirable in terms of characteristics; for instance, what is the impact of the education level/sex/age in your results?   

For Tables 2-6, could you include some plots/graphs that help to highlight differences or advantages?

For readers, it will be interesting that the Authors discuss and give some improvements to the content of the HIC sections. Could you comment on this into the manuscript?

In the Conclusions, can you include what next from this research work?  

Author Response

We are grateful for the reviewers’ comments which assisted us to improve the quality of the presentation of our manuscript. Our responses are shown below as AU in italics. In addition, edits are highlighted in yellow in the manuscript.

Reviewer 2

In this work, the evaluation of the administration of the Health Improvement Card through a randomized controlled card is presented, where Chinese participants are considered. Results demonstrate (in a time window of three months) that HIC-informed health promotion education can improve people’s lifestyle practices. In general, the manuscript is well-written and the topic is interesting; yet, some issues have to be addressed in order to clarify the contribution and applicability for the obtained results.

  • Why is the time window of three months used? Is it enough to obtain reliable results? Can you provide the evolution of the improvements for the month, i.e., for the first month, the second month, the third month (maybe a trend can be observed)?  

AU: A finite period is required to observe any change in behaviour, whether it is sustained and the concurrent effects of any sustained change. Also, lifestyle practices vary at different timepoints through a lifetime. In the first instance of a preliminary investigation, we chose to commence with a 3-month finite period.

  • Please justify the values chosen in section 2.3. What happens with other values?

AU: Further justification for the estimate of sample size is now included, please refer to ling 96 to 103.

  • In the analyzed database, 171 individuals (39 men and 132 women) are considered. Can your results be negatively affected due to the disparity of gender?

AU: While there is a large gender disparity in our cohort, there was no significant interaction effect observed between gender and group (gender*group) for the improvement scores in any outcome of the variables. This is now clarified under 3.1. Please refer to page 6, line 222-224.

  • Figure 1 needs a wider description.

AU: Figure 1 description is now revised and includes details for exclusion and participant loss at follow up.  Please refer to page 5, line 209-212 and updated Figure 1 on page 6.

  • In section 3.1, some reductions are for BMI and WC, but the positive impact of such reductions is not discussed.

AU: Although the changes in BMIs and WCs were relatively small, the improvement in our intervention group was positive and statistically significant. Our discussion has been modified to suggest that our health education intervention findings demonstrate a positive trend to improved health. Please refer to page 10, line 301-305.

  • Although general quantitative results are presented, a deeper discussion is desirable in terms of characteristics; for instance, what is the impact of the education level/sex/age in your results?   

AU: This is an important point that should be refined to further the practice of health education. We recognize that gender, age and education may impact the use and outcomes of using the HIC. However, in-depth examination of the effect of these variables was beyond the scope of this study. We have now addressed this under the limitations of our study. Please refer to page 12, line 378-380.

  • For Tables 2-6, could you include some plots/graphs that help to highlight differences or advantages?

AU: We have considered what graphics might be warranted to elucidate our findings. However, each table contains a considerable amount of information which we believe would generate overly complex plots which might confuse, rather than assist, the readership.

  • For readers, it will be interesting that the Authors discuss and give some improvements to the content of the HIC sections. Could you comment on this into the manuscript?

AU: We have included a paragraph discussing the benefits and the potential clinical utility of the HIC as well as considerations for improvement of its content. Please refer to ‘4.2. Utility of the HIC and Future Studies’ at page 12, line 387-395. However, the HIC was purpose designed into 2 pages to facilitate clinician completion and encourage regular application. A longer more complex version could be considered when clinicians become more experienced with its use.

  • In the Conclusions, can you include what next from this research work?  

AU: The steps going forward from this research are now expanded in paragraph 4.2.

Reviewer 3 Report

IJERPH-912945 A randomised controlled trial to evaluate the administration of the health improvement card as a health promotion tool: A Physiotherapist-led Community-based Initiative.

This submission describes a randomized controlled trial of the Health Improvement Card (HIC) to improve physical and dietary behaviors for non-communicable disease risk reduction among an older Chinese community-dwelling population. The authors present the information clearly with appropriate tables and figures. Minor review for grammatical correction is needed. This comprehensive manuscript will be a valuable contribution to the literature.

Results

Slight wording changes may clarify some statements. For Table 3,  “Subgroup analysis showed that the number of participants in the HIC-intervention group with levels of PA associated with low NCD risk increased 32.3% (55.6% to 87.8%), while the number meeting the healthy diet criterion increased by 20%.”

Author Response

We are grateful for the reviewers’ comments which assisted us to improve the quality of the presentation of our manuscript. Our responses are shown below as AU in italics. In addition, edits are highlighted in yellow in the manuscript.

Reviewer 3

This submission describes a randomized controlled trial of the Health Improvement Card (HIC) to improve physical and dietary behaviors for non-communicable disease risk factor reduction among an older Chinese community-dwelling population. The authors present the information clearly with appropriate tables and figures. Minor review for grammatical correction is needed. This comprehensive manuscript will be a valuable contribution to the literature.

  • Results

Slight wording changes may clarify some statements. For Table 3. ‘Subgroup analysis showed that the number of participants in the HIC-intervention group with levels of PA associated with low NCD risk increased 32.3% (55.6% to 87.8%) while the number meeting the healthy diet criterion increased by 20%.’

AU: Thank you for pointing this out. Table 4 (not Table 3) explains the increase in number of participants in the low-risk categories of diet and exercise. To avoid confusion, we have now moved the subgroup analysis description to a new paragraph after Table 3 and have modified the text. Please refer to page 7, line 235-240.

Reviewer 4 Report

I appreciate the opportunity to review this interesting manuscript entitled “A Randomised Controlled Trial to Evaluate the Administration of the Health Improvement Card as a Health Promotion Tool: A Physiotherapist-led Community-based Initiative”. This is a new approach to assess the benefits of the health improvement card (HIC). I only remark some issues (most of them in methods) in order to improve the quality of this manuscript.

The abstract is clear. Introduction was well structured and correct, but the idea in line 74, about the students, should be place in methods.

At the methods section, there are some questions that should be review. If one exclusion criteria was “known mental illness” was the basal mental state assessed? About the dependent variables, how was the fasting blood sugar, total cholesterol and blood pressures recorded? Was there a nurse in the research team? And what was the intervention with the control group? How was the follow-up?

Results were clear but there is a big difference between sexes of the participants, is there some explanation? It is a great value to add the interval confident to the variable analysis. Discussion: Summarize and explain in a good way how these findings could be beneficial to assess the benefits of HIC. Moreover, It would be interesting to discuss if why the students do not understand when using the HIC may not be appropriate for a particular patient (only 30% strongly agree). One example of physiotherapist advice given to some patient could be added. I miss two lines about future research lines at the end of the discussion.

Conclusions were correct.

Author Response

We are grateful for the reviewers’ comments which assisted us to improve the quality of the presentation of our manuscript. Our responses are shown below as AU in italics. In addition, edits are highlighted in yellow in the manuscript.

Reviewer 4

I appreciate the opportunity to review this interesting manuscript entitled (A Randomized Controlled Trial to Evaluate the Administration of the Health Improvement Card as a Health Promotion Tool: A Physiotherapy-led Community-based Initiative. This is a new approach to assess the benefits of this health improvement card (HIC). I only remark some issues (most of them in methods) in order to improve the quality of this manuscript.

  • The abstract is clear. Introduction was well structured and correct, but the idea in line 74, about the students, should be placed in methods.

AU: Thank you for alerting us to the misplacement of this sentence. The sentence is now removed from the Introduction paragraph.

  • At the methods section, there are some questions that should be review. If one exclusion criteria was ‘known mental illness’ was the basal mental state assessed?

AU: This assessment was based on self-reporting. This is now clarified under 2.2. Participants. Lines 90-92.

  • About the dependent variables, how was the fasting blood sugar, total cholesterol and blood pressure recorded? Was there a nurse in the research team?

AU: The same qualified nurse, blinded to participant group allocation, was invited to assist with recording the biometric data for all participants at every community centre. Physiotherapists are not certified to extract blood samples. All measurements were repeated at the end of the 3-month period. This is now clarified under 2.5 Dependent Variables/ Line 119-121.

  • And what was the intervention with the control group? How was the follow-up?

AU: There was no intervention in the control group apart from issuing a standard brochure on healthy lifestyle practices. We have included an extra sentence to clarify this, please refer to page 4, line 164-165.

  • Result were clear but there is a big difference between sexes of the participants. Is there some explanation?

AU: It appears clear that retired females in China are more active in ‘social’ activities such as grocery shopping, babysitting grandchildren, ‘square dancing’ and are the more active participants in the community centres. We have conducted a factorial ANOVA for the outcome measures with group and sex as independent variables. The gender and interaction effects of group*gender for all outcome measures were not statistically significant, implying that the skewed sex distribution was not an obvious confounding factor in data analysis.

  • It is a great value to add the interval confident to the variable analysis.

AU: Thank you. While we reported the 95% Confidence Intervals (CI) in the analysis of the primary variables, we have now also included the CI in Table 4, illustrating the percentage change in the number of participants in different risk categories. Please refer to Table 4 on page 8.

  • Discussion: Summarize and explain in a good way how these findings could be beneficial to assess the benefits of HIC.

AU: We have included an extra paragraph to discuss the benefits and the potential clinical utility of the HIC. Please refer to 4.2. Utility of the HIC and Future Studies on page 12, line 386-401.

  • Moreover, it would be interesting to discuss if why the students to not understand when using the HIC may not be appropriate for a particular patient (only 30% strongly agree).

AU: Feedback from our students showed that 30% of them strongly agreed and 70% agreed (that is 100% agreed) they understood that ‘the HIC may not be appropriate for a particular patient’ (Table 7).  We are pleased with this response.

  • One example of physiotherapist advice given to some patient could be added.

AU: Examples on dietary and exercise advice are now included. Please refer to page 4, lines 145-149, and lines 153 to 157.

  • I miss two lines about future research lines at the end of the discussion.

AU: This is now included in the additional paragraph under 4.2. Utility of the HIC and Future Studies, page 12, line 386-401.

Round 2

Reviewer 2 Report

All my comments and suggestions have been properly addressed. This Reviewer recommends the manuscript acceptance.